# Study on Particle Manipulation in a Metal Internal Channel under Acoustic Levitation

**DOI:** 10.3390/mi13010018

**Published:** 2021-12-24

**Authors:** Yaxing Wang, Liqun Wu, Yajing Wang

**Affiliations:** School of Mechanical Engineering, Hangzhou Dianzi University, Hangzhou 310018, China; wuliqun@hdu.edu.cn (L.W.); wyj18855034563@163.com (Y.W.)

**Keywords:** metal, internal channel, particle, acoustic levitation and movements, internal machining

## Abstract

In order to study the acoustic levitation and manipulation of micro-particles in the heterogeneous structures inside metal, a test system for internal levitation in three-dimensional space is designed, establishing the 3D motion model of ultrasonic levitation and manipulation of micro-particles. The relationship between levitation force, particle diameter, internal channel size, and transmission thickness is established through the motion manipulation tests of multi-configuration channel levitation micro-particles in components. The results show that the proposed method can realize the following movement of levitation micro-particles at a higher speed and the control of motion accuracy in three-dimensional space. The micro-particles can be reliably suspended and continuously moved inside the components along a predesigned motion trajectory. The results provide an effective and feasible processing scheme for direct processing through the internal spatial structure.

## 1. Introduction

In 2003, Professor Hanmin Shi, a Chinese scholar, created the ‘internal processing’ method by sending processing energy to the interior of a billet, directly processing the internal structure of the parts and manufacturing a new surface within the internal structure of the metal parts [1]. The key to building a new method of ‘internal machining’ is working out how to send the machining energy to the parts. At present, the main method used in internal processing technology consists of a laser for processing transparent material PMMA [2], semiconductor material silicon [3], and HIFU ultrasonic used to focus soundwaves into the organism internal tissue [4]. The application of the above method is based on the matching of energy and material properties. This method has many advantages, such as the low cost of the internal processing equipment and a flexible design that adapts to different processing energy according to actual needs [5]. This paper attempts to study and propose a direct processing method for the internal structure of the metal materials based on acoustic levitation.

Acoustic levitation is an important containerless processing technology [6]. In 1886, Kundt et al. reported the interesting phenomenon of dust particles forming annular patterns in a glass tube placed under standing waves [7]. In 1934, in accordance with the formula of acoustic radiation pressure, King solved the formula of acoustic radiation pressure on a rigid sphere in a plane standing wave field [8]. In 1956, Yosioka’s theory of acoustic radiation force was directly quoted [9]. In 2001, Hancock confirmed that the net radiation force from the standing wave pressure field tends to band the microbubbles at pressure antinodes [10]. In 2006, Professor Wenjun Xie et al. established the single-axis standing wave acoustic levitation test method. According to the principle of ultrasonic standing wave field formation, the diameter of a suspended particle sphere in the standing wave field should be less than half the wavelength [11]. In 2014, the theory of acoustic radiation force was proposed by Takayuki et al. [12]. Stability is a key factor in acoustic levitation [13]. In 2015, Foresti D et al. presented an in-depth analysis of particle levitation stability and the role of the radial and axial forces [14]. Containerless sample environments (levitation) are useful for the study of nucleation, supercooling, and vitrification and for the synthesis of new materials, often with non-equilibrium structures [15]. Containerless treatment of a Ga hypermonotectic alloy is successfully performed with an acoustic levitation technique, indicating that the deformation degree of the reflecting surface showed a direct proportion of the acoustic radiation power. Hong also improved the stability of the single-axis acoustic levitator, especially in the case of levitating high-density and high-temperature samples [16,17,18]. From 2015 to 2021, Asier Marzo proposed an open phase-controlled acoustic levitation array suspension system, which can suspend and manipulate objects greater than half-wavelength of the acoustics wavelength [19,20,21]. In 2018, Zang et al. realized the opening and closing manipulation of the liquid particle shell by using acoustic suspension, and they proposed a new technology for preparing bubbles by transforming droplets into bubbles. When the cavity reaches the critical volume, acoustic resonance induces buckling instability, and the volume of bubbles, induced by acoustic standing wave field, increases sharply, resulting in the sharp expansion and rapid closure of the cavity to form bubbles [22,23]. Under the condition of acoustic suspension, the mechanism of liquid film buckling instability provides a new manufacturing method for the transformation of droplets into large bubbles. Acoustic suspension containerless technology has played an important role in the research of new materials and new microstructures. In recent years, there have also been attempts to study the manufacturing mechanism of internal structures.

However, there are a few reports on the scheme design and feasibility study of the controllable system for controlling the internal processing energy inside the non-uniform space environment of the billet. The purpose of this paper is to study the particle suspension method based on ultrasonics in the non-uniform environment of the internal space of the metal blank and to use the three-dimensional ultrasonic numerical control device to study the transmission of machining energy or machining tools to the internal structure of the metal. In this paper, the technology scheme of internal microstructure particle manipulation and the motion controllable performance of particles in the direct machining of opaque solid materials, such as metals, are established and analyzed. Using the numerical control (NC) device to accurately locate the ultrasonic standing wave field and improve the particle movement accuracy and response speed will greatly improve the possibility of implementing the machining scheme of the metal internal structure. This method has great potential research prospects.

## 2. Mathematical Model and Methods

Figure 1a is the schematic diagram of the uniaxial acoustic levitation device. The ultrasonic wave is transmitted to the processing material through the emission end; it is then transmitted to the inner channel through the processing material. The particles are suspended and manipulated by the cavity standing wave field formed by the emission and reflection of the space inside the cavity of the inner channel. The cavity standing wave field model of ultrasonic propagation is shown in Figure 1b, in which h is the height of the channel.

### 2.1. Acoustic Levitation Force and Motion Model of Particles

It is assumed that the characteristics of the particle material are isotropic and that the deformation generated by the force is uniform in all directions. The environment is a continuous ideal medium, and the force process is adiabatic. At the same time, the ultrasonic wave is a small amplitude, and the ambient temperature is room temperature. According to King’s theory, the general expression of acoustic pressure is established as follows:(1)$$\Delta p=\rho_{m}\varphi +\frac{1}{2}\frac{\rho_{m}}{c^{2}}\varphi^{2}-\frac{1}{2}\rho_{m}u_{p}^{2}$$
where *p* is the acoustic pressure; Δ*p* is the acoustic pressure variation; *ρ_m_* is the medium density; *φ* is the speed potential; c is the propagation velocity of the ultrasonic wave in the medium; *u_p_* is the particle velocity.

When the particle is in the acoustic field, the radiation force of the particle can be obtained by combining the corresponding natural boundary conditions. The radiation force in the standing wave direction can be expressed as follows:(2)$$F_{z}=-\frac{\pi p_{0}^{2}V_{p}\beta_{m}}{2\lambda }\varphi (\beta ,\rho )\sin(2kz)\;\varphi (\beta ,\rho )=\frac{5\rho_{p}-2\rho_{m}}{2\rho_{p}+\rho_{m}}-\frac{\beta_{p}}{\beta_{m}}\;k=n\pi \;n\in N$$
where *λ* is the wavelength; *p*_0_ is the acoustic pressure at the origin of the coordinate; *z* is displacement; *V_p_* is the particle volume; *ρ_p_* is the particle density; *β_p_* is the particle compressibility coefficient; *β_m_* is the compressibility coefficient of the medium.

The compressibility coefficient is related to the wave velocity in the medium, namely *β* = 1/(*ρc*^2^). If the acoustic ratio factor *ϕ* (*β*,*ρ*) is positive, the restoring force of the particle at the standing wave node is shown in Figure 2a,b shows the horizontal motion force model of particles. In the vertical direction, the particles are affected by acoustic radiation force *F_z_*. In the horizontal direction, particles are also affected by radiation forces *F_x_* and *F_y_*.

According to the theory of acoustic radiation force proposed by Takayuki et al., the radiation force in the horizontal X direction can be expressed as follows:(3)$$F_{x}=\frac{6k{\bar{p}}^{2}V_{p}}{\rho_{p}c^{2}}\left[\frac{\sin(2kx)\cos(2kx)}{{\left(2kx\right)}^{2}}-\frac{\sin^{2}(2kx)}{{\left(2kx\right)}^{3}}\right]$$
where $\bar{p}$ is the effective acoustic pressure, *T* is the periodic.

When the particle is accelerated in the fluid medium, the velocity of the particle will gradually increase, and the velocity of the surrounding fluid will gradually increase, this force can be expressed by virtual mass force:(4)$$F_{um}=-\frac{1}{2}\rho_{m}V_{p}\frac{du_{r}}{dt}$$
where *u_r_* is the relative velocity.

The resistance of the object to relative motion in the fluid is called viscous resistance. The expression of viscous resistance is established as follows:(5)$$F_{v}=-6\pi \eta ru_{r}$$
where *η* is the medium viscosity, and *r* is the particle radius.

In the vertical direction, the motion equation can be expressed as follows:(6)$$m_{p}\frac{d^{2}z}{dt^{2}}=F_{z}+F_{v}+F_{um}+G\;m_{p}\frac{d^{2}z}{dt^{2}}=-\frac{\pi p_{0}^{2}V_{p}\beta_{m}}{2\lambda }\varphi (\beta ,\rho )\sin(2kz)-6\pi \eta ru_{rz}-\frac{1}{2}\rho_{m}V_{p}\frac{du_{rz}}{dt}-\rho_{p}V_{p}g$$
where *m_p_* is particle mass; *G* is the gravity of particles; *g* is Gravity acceleration.

The horizontal motion equation of suspended particles can be expressed as follows:(7)$$m_{p}\frac{d^{2}x}{dt^{2}}=F_{x}+F_{v}+F_{um}\;m_{p}\frac{d^{2}x}{dt^{2}}=\frac{6k{\bar{p}}^{2}V_{p}}{\rho_{p}c^{2}}\left[\frac{\sin(2kx)\cos(2kx)}{{\left(2kx\right)}^{2}}-\frac{\sin^{2}(2kx)}{{\left(2kx\right)}^{3}}\right]-6\pi \eta ru_{rx}-\frac{1}{2}\rho_{m}V_{p}\frac{du_{rx}}{dt}$$

Due to symmetry, the force in the Y-axis motion direction is consistent with the force in the X-axis direction. In summary, by combining Equations (6) and (7), the motion model of a particle in a standing wave field is established as follows:(8)$$\begin{array}{l} m_{p}\frac{d^{2}x}{dt^{2}}=F_{x}+F_{vx}+F_{umx}\\ m_{p}\frac{d^{2}y}{dt^{2}}=F_{y}+F_{vy}+F_{umy}\\ m_{p}\frac{d^{2}z}{dt^{2}}=F_{z}+F_{vz}+F_{umz}+G \end{array}\}$$

### 2.2. Analysis of the Reflection Coefficient and Transmission Coefficient of the Vertical Incident Wave

When the ultrasonic wave enters the new interface of the material, part of the energy penetrates the interface of the material, while the other part of the energy reflects back on the interface of the material. The incident, transmission, and reflection waves are shown in Figure 3. The incident stress is *σ_I_*, the reflection stress is *σ_R_*, the transmission stress is *σ_T_*, and the acoustic wave propagation is limited to the horizontal X-direction.

Because the acoustic energy is proportional to the square of the pressure, the energy transmission distribution formula and the energy reflection distribution formula of the material interface can be obtained. The energy per unit time and per unit area through the direction perpendicular to the ultrasonic direction is called the energy density, namely the wave intensity, which is expressed by *I_I_*, *I_R_*, and *I_T_*:(9)$$\begin{array}{l} \frac{I_{R}}{I_{I}}={\left(\frac{B-1}{B+1}\right)}^{2}\\ \frac{I_{T}}{I_{I}}=\frac{4B}{{\left(B+1\right)}^{2}} \end{array}\}\;B=W_{2}/W_{1}$$

The ultrasonic wave velocity of several common materials is shown in Table 1. The acoustic impedance of the material can be calculated according to Equation (9) and Table 1.

An ultrasonic wave is emitted from the transmitter of the transducer and transmitted to the processing material, which will cause some reflection loss. Considering the internal processing environment, it is necessary to obtain higher transmission efficiency at the first interface and higher reflection efficiency at the second interface. Aluminum is selected as the processing object material, and 45 steel is selected for the ultrasonic horn material. The transmission path of the design wave is steel–aluminum–air–aluminum–steel.

(1) Firstly, the reflection coefficient, transmission coefficient, reflection ratio, and transmission ratio of steel–aluminum during ultrasonic conduction are calculated, and the results are shown in Table 2.

(2) The reflection coefficient, transmission coefficient, reflection ratio, and transmission ratio of aluminum–air in ultrasonic transmission process are calculated, and the results are shown in Table 3.

From the calculation results shown in Table 2 and Table 3, it can be seen that the acoustic transmission path design of steel–aluminum–air–aluminum–steel meets the design requirements of ultrasonic energy transfer and energy distribution in each link.

### 2.3. Acoustic Levitation Control Test System Model and Design Implementation

Similar to the principle of the single-axis acoustic levitation device, an ultrasonic standing wave three-axis acoustic levitation device is constructed by using three groups of transmitters and reflectors, as shown in Figure 4a. The ultrasonic standing wave field is the spatial standing wave suspension field where the standing wave nodes intersect due to the spatial vertical crossing and staggering. The space formed after pushing the half-wavelength distance from the surface of the component to the inside of the component is the controllable range of motion. Any position of the internal point can become a column of standing wave nodes in the three-dimensional orthogonally distributed standing wave field, namely, the suspended potential well array. The three-axis acoustic levitation device in the component is composed of an ultrasonic transmitting reflection end, a standing wave levitation field, a rotating bracket, and a rotating and translational driving device of the transmitting end, etc. After repeated optimization design, the movable standing wave levitation field motion control scheme shown in Figure 4b is constructed.

Three ultrasonic generator parameters are the same; frequency is f = 30 kHz, power is *p* = 300 W, and amplitude is A = 30 μm. Zolix’s TSA200-E ball screw guide rail is selected as the motion device of the three-dimensional acoustic levitation system. Each guide rail is responsible for a direction of freedom. The ultrasonic transmitter is fixed on the slider with a fixed bracket. Three numerical control devices are used to control the stepper motor to drive the ultrasonic transmitter to complete three-dimensional space motion. Each guide rail can be loaded to the stepper motor by a manual or numerical control device to make it move. The stepper motor model is 42BYG250A. The actual test device is composed of a platform base, fixed support, rotary bracket, rotary launch block, movable device, and base block, as shown in Figure 5.

## 3. Results

### 3.1. Experiment on the Relationship between Particle Suspension and Transmission Thickness

The processing material used in the test is aluminum; the size is 15 mm × 15 mm, 20 mm × 20 mm, 25 mm × 25 mm, 40 mm × 40 mm, and 50 mm × 50 mm, and the height is 50 mm and 100 mm. The selected suspended particles are polyethylene foam particles with a density of 0.015~0.03 g/cm^3^. Air wave velocity is v = 340 m/s, ultrasonic generator frequency is f = 30 kHz, and the ultrasonic wavelength is almost 10 mm.

Taking into account the theoretical formula, the power of the generator and the propagation performance of the processed material, experiments with different transmission thicknesses were designed. Figure 6a shows the suspension effect of foam particles with a diameter of 1 mm when the transmission thickness of the internal processing component is H = 16 mm and the diameter of the channel hole is 5 mm. Figure 6b is an acoustic field analysis of the particle suspension effect in the circular channel. After a series of transmission thickness suspension tests, the relationship between the acoustic levitation force and transmission thickness can be obtained. When the transmission thickness of the internal processing component is within 16 mm, the particles can be stably suspended, and the effective transmission thickness is about 1.5 times the ultrasonic wavelength.

### 3.2. Experiment on the Relationship of Particle Suspension with Channel Diameter and Particle Diameter

Two circular channels of 5 mm and 10 mm and three long circular channels of 10 mm × 25 mm, 10 mm × 40 mm, and 10 mm × 45 mm were designed, respectively. Figure 7a,b shows the long circular channel of 10 mm. The foam balls with diameters of 1 mm, 2 mm, 3 mm, and 4 mm were selected for experimental verification to determine the optimal particle diameter. In the experiment, when the particle diameter is within the range of 0 to 1 mm, the particle suspension stability increases with the increase in the particle diameter. When the particle diameter is within the range of 1 to 4 mm, the particle stability does not change significantly and is in a relatively stable state; when the particle diameter is greater than 4 mm, the particle rapidly loses stability and cannot suspend stably.

Figure 7a,b shows horizontal and vertical suspension effects in the long circular channel, Figure 7c shows the particles’ follow motion and local acoustic field. Figure 7d shows the suspension stability test results of 2 to 20 mm foam balls. The experimental results show that: (1) the particle suspension stability of the 10 mm round channel or the 10 mm × 25 mm, 10 mm × 40 mm, and 10 mm × 45 mm oblong channel is better than that of the 5 mm round channel; (2) the diameter of the particles cannot be greater than half-wavelength (5 mm) and need to be limited to half wavelength; (3) the particles’ follow motions are consistent with the model in Equation (8) of the Section 2, as the transducer moves and the wave node moves, the force of *F_x_* and *F_y_* changes and the particles follow.

### 3.3. Particle Space Arbitrary Point Suspension and Suspension Displacement Motion Test

The 2 mm foam ball and the 10 mm × 25 mm, 10 mm × 40 mm, and 10 mm × 45 mm oblong channels are used to design the arbitrary point suspension and motion following test. Figure 8a,b shows the displacement following test. In the test, the launch end is moved to a certain distance in the positive direction of X, and the small ball in the inner channel is also moved to the corresponding distance. After a series of micro-displacement change tests, the displacement of the ultrasonic device and the displacement of the particle are compared, as shown in the table in Figure 8c.

The test results show that the particle displacement follows a fast response, and the error of particle displacement following motion is within ±0.71%. The particle displacement accuracy in the inner channel is high. Combined with the results of any point test and the analysis of displacement motion following accuracy, it can be seen that the suspended particles can follow the motion of the three-dimensional orthogonal nodes of the ultrasonic standing wave. The acoustic levitation control platform has reached the basic requirements of motion control, which lays a foundation for the design of a particle motion trajectory control model, motion accuracy analysis, a machining accuracy test, and internal channel machining research.

### 3.4. Rotational Motion Test of Particles

Experiment 1: Different numbers of particle rotation tests

According to the theoretical analysis of the Section 2, the particles in the one-dimensional standing wave field are affected by the ultrasonic radiation force, which makes the particles gather to the node position. Figure 9 shows the suspension state of different numbers of particles at the node position.

Different numbers of particles rotate under the action of the driving force; the stability of the movement and the maximum speed of the movement are inevitably different. In order to study the state of the most stable circular motion of the particles, the rotating motion experiments of different numbers of particles are designed. Figure 9 is the rotation test of different particles. The maximum speed data of different numbers of particles measured by a high-speed camera are shown in Figure 10.

It can be seen from Figure 10 that with the increase in the number of particles at the node position, the maximum speed of the stable rotation of particles increases at first and then decreases. Due to the increase in the number and the irregular shape of multi-particle aggregation, the size of the force arm generated by the driving force is different. When the number of particles is 3, the particle can reach the maximum speed when rotating, and its stability is the best.

Experiment 2: Particle rotation inside metal

A single particle can be stably suspended in a solid medium and can follow the standing wave field horizontally. The rotational motion experiment of suspended particles in a solid medium under driving radiation force is carried out. In accordance with the above test, three particles are suspended in the solid channel to prepare for the rotation movement, and then the other surface of the component is applied to drive the radiation force emitter to drive the particle rotation, as shown in Figure 11.

In summary, the velocity of particles following the standing wave field is related to the frequency of the standing wave field. The higher the frequency is, the higher the maximum velocity is. With the increase in the standing wave field distance, the velocity of the following motion decreases gradually, and when the distance is at half-wavelength integer times, the velocity is the largest relative to the adjacent distance. The most stable state of particle rotation is found when the number of particles is three, the reflection distance of the standing wave field is a wavelength of the standing wave ultrasonic, and the distance between the axis of the horn at the driving radiation force transmitter and the center of the node is half of the wavelength of the driving ultrasonic.

## 4. Conclusions

(1) Particles can achieve stable acoustic levitation in the metal material component’s internal space. The basic space includes a basic spherical space, horizontal channel, vertical channel, and right-angle channel, etc. Some particles’ suspension performance under acoustics standing waves concludes that it relates to component shape, transmission thickness, channel size, particle size, acoustic wavelength, and ultrasonic power. 

(2) Small particles can be acoustically manipulated in the metal internal space by the external acoustics field. Suspended particles can move in a larger range, following a quick response at high moving accuracy with micro-vibrations. The suspended particles can also be driven to rotate, and the rotational speed reaches its maximum when the particle’s radius is equivalent to half the acoustics wavelength; the particle rotation speed is related to the distance between the particle and the driving emission end face. With the increase in distance, the speed increases first and then decreases.

(3) Particles can be driven along the preset trajectory in the 3D metal internal space beyond the critical space range of half-wavelength distance between the internal surfaces. In this range, the motion of the particles is controllable, and the particles follow the motion by following the acoustics field of the three-dimensional orthogonal standing wave node. Particles can be driven to any point in the channel by NC to control the movement of the three-dimensional positive energy exchanger and to form a preset arbitrary motion track in the internal space of the component.

## Figures and Tables

**Figure 1 micromachines-13-00018-f001:**
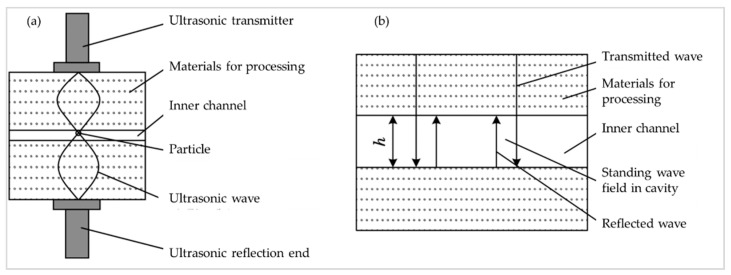
(**a**) Diagram of the single-axis acoustic levitation device; (**b**) Diagram of the formation model of a standing wave in the cavity.

**Figure 2 micromachines-13-00018-f002:**
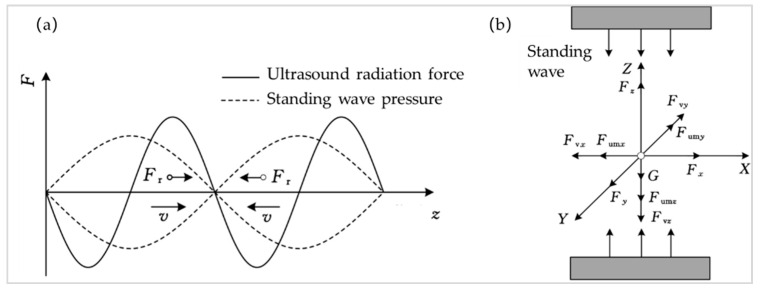
(**a**) The force diagram of particles in a one-dimensional ultrasonic standing wave field; (**b**) Diagram of the Force Model of Particle Horizontal Motion.

**Figure 3 micromachines-13-00018-f003:**
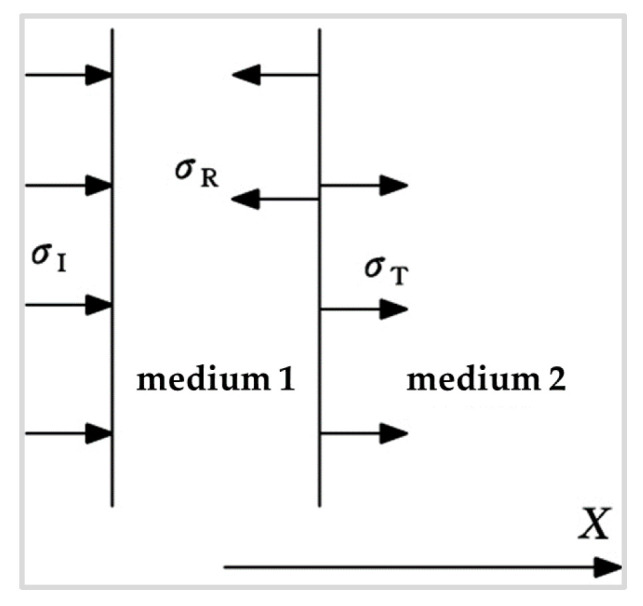
Incidence, reflection, and transmission.

**Figure 4 micromachines-13-00018-f004:**
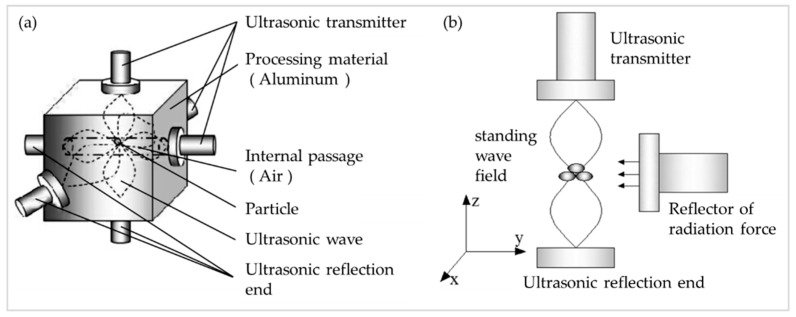
(**a**) Diagram of the solid medium triaxle acoustic levitation device; (**b**) Diagram of the schematic design of the acoustic levitation control system.

**Figure 5 micromachines-13-00018-f005:**
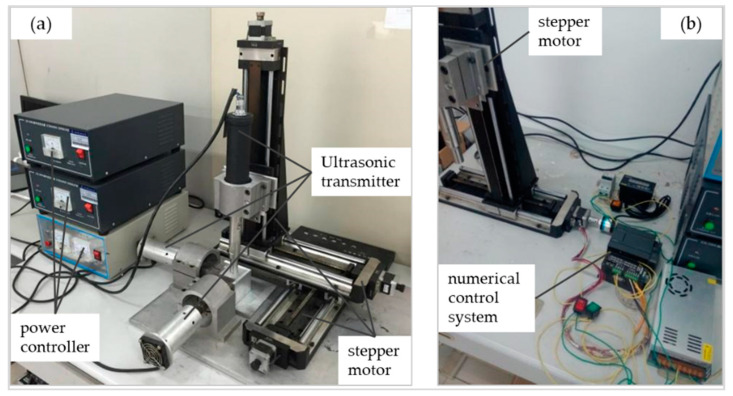
(**a**) Diagram of the three-dimensional acoustic levitation system; (**b**) Diagram of the numerical control system, which controls the stepper motor in (**a**).

**Figure 6 micromachines-13-00018-f006:**
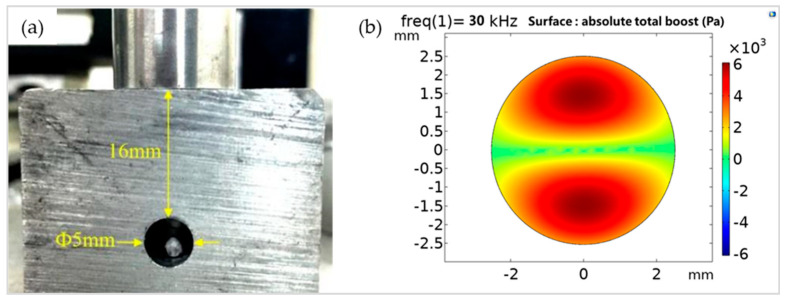
(**a**) Diagram of the suspension effect at H = 16 mm; (**b**) Diagram of the acoustic field analysis of the particle suspension effect in the circular channel.

**Figure 7 micromachines-13-00018-f007:**
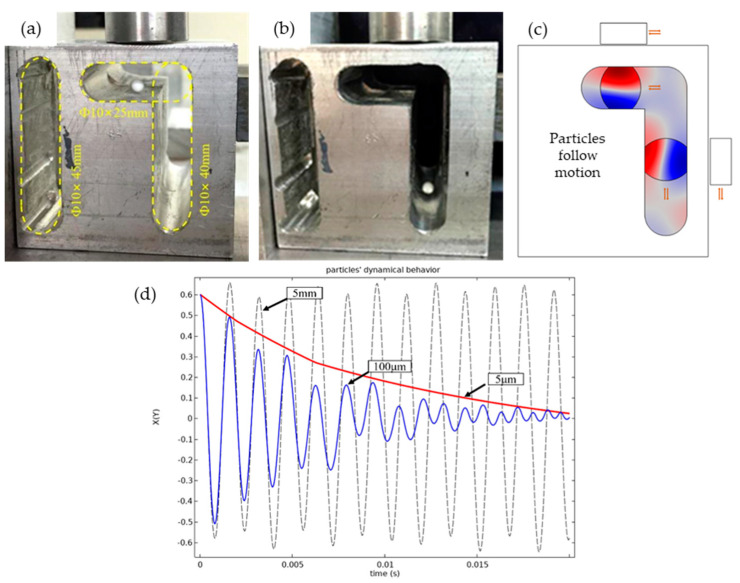
(**a**,**b**) Diagram of the horizontal and vertical suspension effects in the long circular channel; (**c**) Diagram of particles’ follow motion and local acoustic field; (**d**) Diagram of the suspension stability test results of 2 to 20 mm foam balls.

**Figure 8 micromachines-13-00018-f008:**
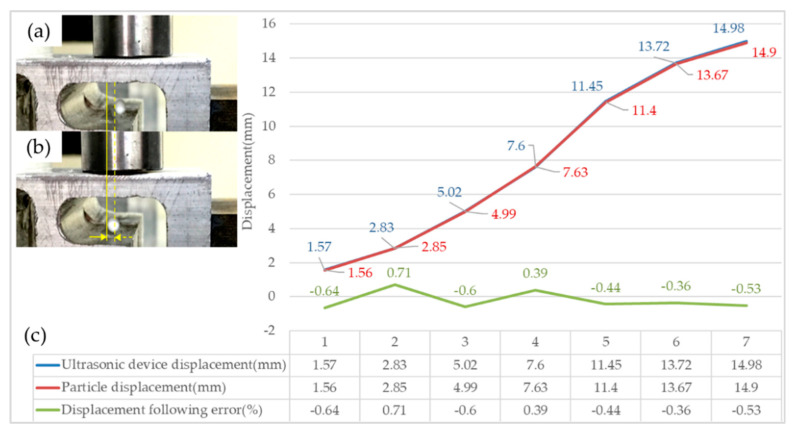
Particle displacement motion: (**a**) Diagram of the initial position; (**b**) Diagram of the micro-displacement change; (**c**) Diagram of the table of the particle displacement following error.

**Figure 9 micromachines-13-00018-f009:**
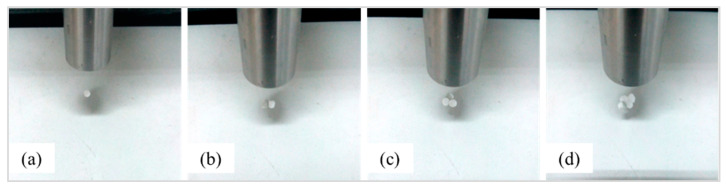
The suspension state of different numbers of particles at the node position: (**a**–**d**) Diagram of different numbers of particles.

**Figure 10 micromachines-13-00018-f010:**
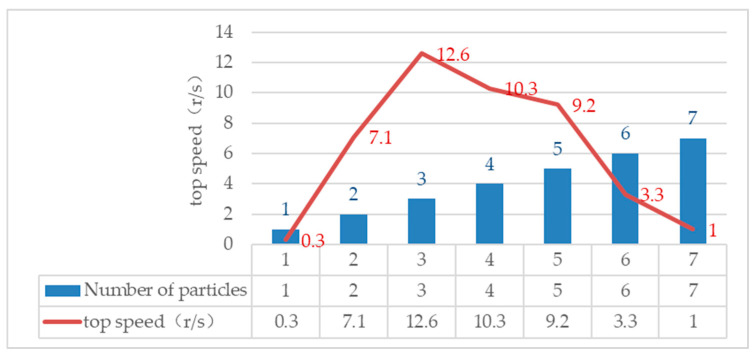
Number of particles and maximum speed.

**Figure 11 micromachines-13-00018-f011:**
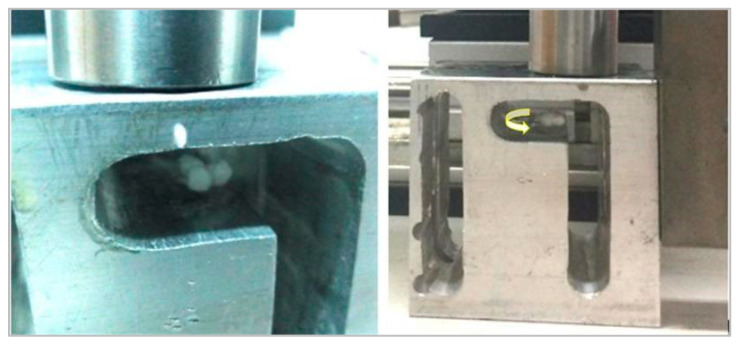
Particle rotation inside metal structure.

**Table 1 micromachines-13-00018-t001:** Wave velocity of several common materials. Acoustic velocity meters of common materials measure longitudinal acoustic velocity, so we only calculate the longitudinal acoustic data (“Vertical” in Table 1).

Material	Density (g/cm^3^)	Wave Velocity (m/s)	
		Vertical	Horizontal
Air	0.001	340	/
Aluminum	2.7	6250	3100
Steel	7.8	5850	3230
Polyethylene	0.920	2000	/

**Table 2 micromachines-13-00018-t002:** Acoustic conduction reflection ratio and transmission ratio of the steel–aluminum interface.

Reflective Interface	*σ_R_/σ_I_*	*I_R_/I_I_*	*σ_T_/σ_I_*	*I_T_/I_I_*
**Steel–Aluminum**	−0.46	0.212	1.46	0.788
**Aluminum–Steel**	0.46	0.212	0.54	0.788

**Table 3 micromachines-13-00018-t003:** Acoustic reflection and transmission ratio of the aluminum–air interface.

Reflective Interface	*σ_R_/σ_I_*	*I_R_/I_I_*	*σ_T_/σ_I_*	*I_T_/I_I_*
**Aluminum–Air**	−0.99996	0.999922	1.99996	0.000078
**Air–Aluminum**	0.99996	0.999922	0.00004	0.000078

## Data Availability

Not applicable.

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
