# Peer review of "Study on Particle Manipulation in a Metal Internal Channel under Acoustic Levitation"

_micromachines, 2021, doi:10.3390/mi13010018_

Round 1
Reviewer 1 Report
In this study, the author shows the particle manipulation in the heterogeneous channel. the particle levitation and manipulation in the circular channel were performed at various experimental conditions, such as channel geometry and particle size. Although the concept and phenomena are interesting and are suitable for Micromachine, the theoretical evidence was not shown in this study based on the described in the Section 2.1-2.2. Thus, I do not recommend this particle for the publication in Micromachine.
Comment
- The character used in the explanation of the parameter differs from that in the equation. For example, p in Eq.(3). This should add the overline.
- p.2 L71; what is the "different" sound field?
- p.2 L78; where z is the distance from?
- p.3 L92 Eq. (3), what is the x?
- p.6 L191-199; why does the particle levitation occur in a circular cavity? Eqs. (2) and (3) can be applied to the case of the ultrasound standing wave in a cell sandwiched between flat plates.
- Overall; based on the theoretical model (Eqs. (1)-(9)), the result about the particle levitation should be evaluated.
Reviewer 2 Report
The authors demonstrate a system in which Styrofoam balls are levitated and translated in a machined cavity. While certain aspects of this are interesting and potentially novel, especially the use of a machined resonating cavity to generate an acoustic field, this work is lacking in context and analysis for this novelty to be conveyed effectively. This work would benefit strongly from delineating why their tested geometries are useful and why they were designed this way, how this would or could translate to practical use, illustrating how this is different from previous work, and showing what the acoustic field looks like in the cavity interior. Moreover, little physical insight is presented in terms of why the particles might behave in the acoustic field in they way that they do. The authors are advised to address my comments in re-working their manuscript.
- The introduction is unclear about the potential uses of this technology – there may be some language issues here, but the relevance of the presented work to sending ‘machining energy’ to the internal structure of a part is not obvious. How does moving Styrofoam balls in a cavity with a given shape help this?
- Figure 1-2 and Figure 3-4 could be Figure 1(a,b) and Figure 2(a,b) respectively. The same goes for Figure 6-7.
- In table 1, different sound speeds are listed for different orientations. For isotropic materials such as aluminium (unless this is some kind of non-amorphous metal) and steel, it’s unclear why the wave velocity should be different in different directions. Are these different wavemodes instead (i.e. longitudinal or Rayleigh mode)? If so this can be re-titled appropriately.
- What is the significance of the “L” shaped cavity shown in the experimental results? Why is this interesting or useful? What does or should the acoustic field look like in this system? Could a simulation or model of the generated acoustic field be produced?
- Lines 182-184 look like they were taken from a manuscript template, and should be removed.
- The sound speed in air is approx. 340 m/s. Where does the value of 300 m/s come from (line 189)?
- Line 230 – “Include table of particle displacement following error”. Is this referring to a table in the manuscript? Should this text be here?
- Figure 11 – should the blue line be plotted here as well?
- Why are the particles rotating in Figure 13? What is the physical mechanism underpinning this motion?
- This work would benefit strongly from the inclusion of supplementary information videos showing particle levitation and translation.
- The left channel in figure 10 does not appear to be shown in experimental demonstrations in the figures. Why?
- Lines 213-217 state experimental results, but do not explain why these results might occur. Can these be linked back to the analytical predictions in the equations?
Reviewer 3 Report
Wang et al. presented "Study on particle manipulation in metal internal channel under acoustic levitation". In this work, metal walled internal channels are used to establish standing waves and manipulate particles through acoustic levitation. It is important to understand and apply acoustic manipulation within heterogenous structures. The authors proposes the thick metal walls as the medium that the acoustic waves ravel through before establishing the standing wave within the cavity. Even though the concept is somewhat interesting and the findings are meaningful, the presentation of the manuscript must be improved.
1- All the figures requires editing, restructuring and labelling. For example, fig 1 and 2 can be merged and constitute a) and b) panels of a single figure. Same applied to Figure 2 and 3.
2- Fig 8 should be labeled. the left and the right panels can be labeled as a and b. Besides, ahh the components in the figures should be annotated. Figure captions are needs to be more explanatory.
3- Figure 11 should definitely be edited and improved. Graph and table components of this figure can be labelled as c) and d) as well.
4-The graph in Figure 11 needs to have labelled axes.
5- Fig 13 needs labels in the graph. Please check all the graphs to make sure that they have x and y axes labelled correctly.
6- When explaining the equations, please use the same font as in the equation. For example, when defining the density in equation 1, you should either insert the symbol of density as an equation within the text or change the font to match to the symbol in the equation. check your formatting when writing the particle velocity.
7- Introduction is way too short. The authors should include more relevant literature. References should be increased along with the discussions.
8- In section "3.4", it reads "According to the theoretical analysis of the second chapter," does this refer to the "2. Materials and Methods" section of the manuscript ? It may be better to write "... theoretical analysis of the second section (or section two)".
Round 2
Reviewer 1 Report
The authors revised the manuscript substantially according to my suggestion. Although the experimental results for the levitation coordinate and velocity were not supported based on the theoretical equations (Eqs. (1)-(9)), this manuscript shows the interesting phemonenology. Thus, I agree with the publication of this manuscript for the Micromachines.
Reviewer 3 Report
Authors addressed my concerns and comments in the revised version. Therefore, I recommend acceptance of the manuscript.